# Dynamic Analysis of the Complex Motion of Three-Section Cudgel in Wushu Sports

**Man Xu [1,†], Yiran Jiang [2,†], Xingfu He [3], Juntong Yuan [4] and Ying Gao [5,*]**

1    Public Sports and the Arts Department, Zhejiang University, Hangzhou 310058, China; hzxman@zju.edu.cn
2    Institute of Biophysics, Chinese Academy of Sciences, Beijing 100101, China; jiangyiran21@mails.ucas.ac.cn
3    School of Civil Engineering and Architecture, Zhejiang University, Hangzhou 310058, China; 3160104811@zju.edu.cn
4    Department of Physical Education and Health, Peking Sports University, Beijing 100091, China; 2021010563@bsu.edu.cn
5    Department of Sports Science, College of Education, Zhejiang University, Hangzhou 310058, China
*    Correspondence: yigao@zju.edu.cn
†    Man Xu & Yiran Jiang have equal contribution to this article and serve as co-first-authors.

**Abstract:** Purpose: To provide some suggestions on how to effectively master these movements during training. Methods: The dynamics method and ABAQUS simulation analysis were used to analyze the two technical movements of the three-section cudgels: wrestling cudgels and retrieving cudgels. Results: In the process of wrestling the cudgel, the best effect is achieved when the height of the cudgel holding hand is between 70 cm and 80 cm from the ground. The maximum height of the rebound was very similar with different initial angular velocities, at 4.5–9 cm. The initial angular velocities caused significant impact to the horizontal movement of cudgel at 8 s. By excluding the errors, the horizontal movement of cudgel increased approximately linearly with the increase of the initial angular velocity. Conclusions: When the height between held cudgel and the ground was controlled at 1.5 times the height of the middle section, the rebound of the tail section was the least. When completing the movement of three-section cudgel, the cudgel body should be in the same plane perpendicular to the ground and to better retrieve the three-section cudgel. The main factor affecting the cudgel wrestling was the height between the cudgel holder and the ground.

**Keywords:** three-section cudgel; cudgel technique; competition routine; dynamics; collision theory

## 1. Introduction

The three-section cudgel is a typical soft instrument in Wushu routine. It can be traced back to the "Panlong cudgel" in the Song Dynasty [1,2]. Zhao Kuangyin, the first emperor of the Song Dynasty, changed a rigid cudgel into a three-section cudgel with three rigid bodies, which made the movement of the cudgel more complex and flexible [3–5]. Current Wushu research on dynamics is mainly focused on movements of competitive Wushu jumping, while the investigation on the main movements of an instrument, are rarely involved.

The three-sectioned cudgel routine has a variety of movements, which are compact and coherent. Skilled control will give it full play to the flexibility and exploit its strong power with the characteristics of short hitting and striking long and far. The wrestling, throwing, retrieving, lifting, poking cudgel and other movements in the three-section cudgel set combining with jumping, turning over, rolling and other typical movements of the three-section cudgel make this kind of wushu instrument very powerful but difficult to practice. In the practice of beating three-section cudgel, the practitioners often injured because the cudgel tip bounces back to the practitioner after heavy force; and finishing position, the inaccuracy of the force point cannot receive the three-section cudgel into the starting position [6].

In order to avoid the deducting points due to collision, falling and other phenomena, there are two movements often removed from the practice routine of the three-section cudgel, causing that the three-section cudgel routine movement lost its original characteristics gradually. This study was aimed to using the dynamics method to analyze the mechanical principle of wrestling and retrieving cudgel. Furthermore, the whole process of wrestling action was analyzed by ABAQUS. Based on the actual training experience, the results of theoretical analysis and software simulation were summarized and the key technical points for the practitioners were put forward.

## 2. Materials and Methods

### 2.1. The Instrument of the Three-Section Cudgel

As shown in Figure 1, the three-section cudgel consists of three short cudgels of equal length and chained end to end by small iron hoops. The three short cudgels are named as root section, middle section and tail section, respectively, whose size is 2 cm to 3 cm diameter and 50 cm to 80 cm long. The middle section is relatively thin [7]. The chain is linked by three rings, and each chain is 5 cm long. Compared with other instruments, the three-section cudgel is endowed with more flexibility in the meanwhile keeping its strong rigid body, and its hitting range can be either long or short with plenty of diversifying attack way. The three-section cudgel is flexible, and the ways of attack and grip are also varied. Four main basic grip methods of three-section cudgel are: holding the root section and the tail section; holding the root section and wrestling the middle section and the tail section; holding the root section and the middle section, wrestling the tail section; and holding middle section and wrestling the tail section and the root section. The three-section cudgel moves used in this study were parts of the moves, which included in the traditional apparatus competition sets. The selected moves involve the first, the second and the third holding processing. In this study, the beating and retrieving actions are the second grip, that is, holding the root section, and wrestling the middle section and the tail section.

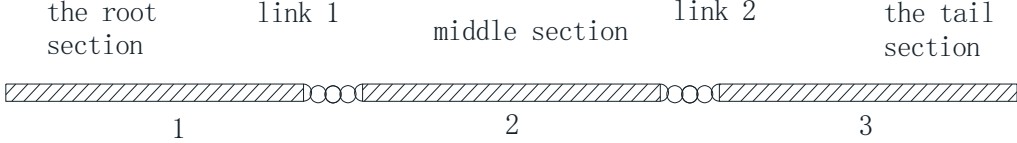

**Figure 1.** Schematic diagram of three-section cudgel model. (1,2,3 refers to the sequence arranges from where cudgel is held to hit).

### 2.2. Dynamic Analysis of Wrestling the Three-Section Cudgel

2.2.1. Mechanical Analysis of Three-Section Cudgel Wrestling Technique

In the three-section cudgel set, there is often the action of wrestling the cudgel. It can be very swift and enable hitting with strong force at the end of the cudgel. However, it is also easy to cause rebound at the end of the cudgel, which tests the trainee's strength and height control ability in the process of wrestling the cudgel (Figure 2). There are certain techniques in the practice of wrestling cudgels. If the skills are not mastered, it will be easy to cause a sharp rebound at the end of the cudgel. It is possible to induce users' injures, which is not only unable to control the opponent but also does harm to the practitioners themselves. Thus, the rebound mechanism of the end of the three-section cudgel is aimed to investigate through the impact theory of mechanics, and the technical essentials of throwing cudgel are analyzing.

The usual three-section cudgel consists of three cudgels and two sets of iron rings. The elastic modulus E of the stick is in the range of 9–12 GPa, Poisson's ratio ranges from 0.16 to 0.18. The elastic modulus E of the ring is about 210 GPa, and the Poisson's ratio is in the range of 0.25 to 0.30 [8]. The premise of rigid body dynamics research is that the stiffness and strength of the structure are enough, which will not have a great influence on the research results. The strength and stiffness of the three-section cudgel are great. In

the process of three-section cudgel wrestling, the external force on the three cudgels has little effect on its internal deformation, and the deformation of the cudgel itself is very small. In this study, the extended displacement movement of three cudgels in the process of wrestling was researched, which will not have a great influence on the research results. Therefore, the relatively insignificant deformation of the three cudgels in the process of wrestling is ignored and the three cudgels are regarded as rigid bodies for dynamic analysis. In addition, in order to simplify the analysis, the impact between the cudgel and the ground is assumed to be an elastic impact, and the interaction between different materials during force transmission is not considered temporarily.

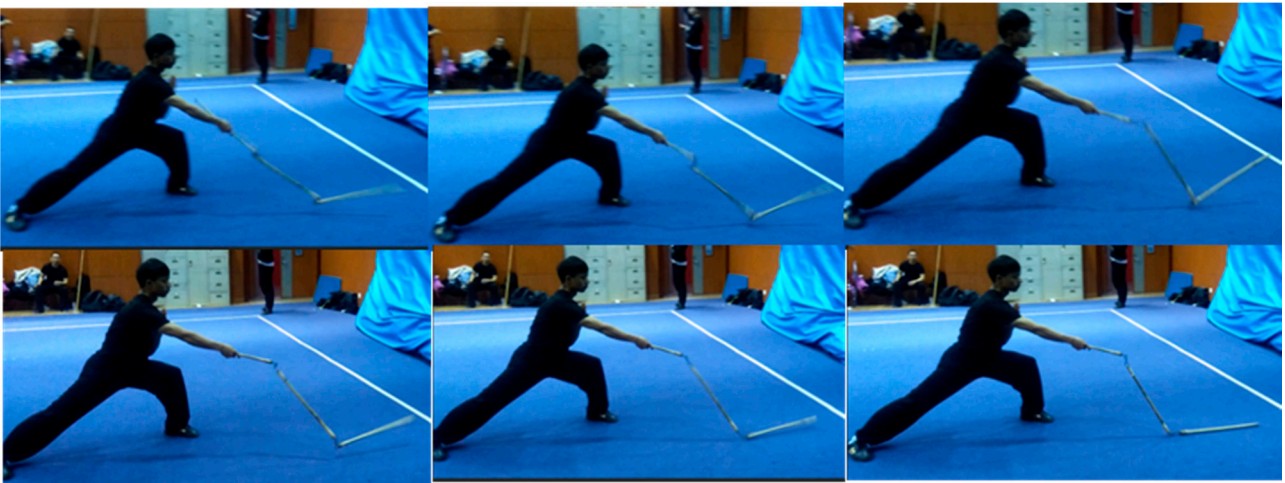

**Figure 2.** The process by which the cudgel bounces after being thrown.

It is assumed that the three sections of the cudgel are homogeneous rods with equal length (*l*), the angle between rod and ground is *θ*, and instantaneous angular velocity before a collision with the ground is *ω*, and m in mass. To simplify the calculation, it is assumed that the ground is smooth (although there is no absolutely smooth ground in reality), as shown in Figure 3. While collision between the cudgel and the carpet or floor, restoration factor is needed to be considered [9]. Based on the consideration that the material of cudgel is generally stiffer and stronger, we can further assume that the recovery factor *e* = 1, which is the ideal case. In other words, it means that the deformation of the rod can be completely recovered at the end of the collision, without loss of kinetic energy, i.e., being a completely elastic collision [10].

Considering the cudgel motion in plane during the collision, the collision equation is then derived from the plane motion of the rigid body [11].

$$\mathrm{m}v'_{Cx} - \mathrm{m}v_{Cx} = \sum I_x \tag{1}$$

$$\mathrm{m}v'_{Cy} - \mathrm{m}v_{Cy} = \sum I_y \tag{2}$$

$v_{Cx}, v'_{Cx}$—The horizontal component of the velocity of the centroid of the tail section cudgel at the moment before it hits the ground and rebounds (i.e., X direction)

$v_{Cy}, v'_{Cy}$—The vertical component of the velocity of the centroid of the tail section cudgel at the moment before it hits the ground and rebounds (i.e., Y direction)

$I_x, I_y$—The impulse of the tail section in the horizontal and vertical directions

$$J_c\omega_2 - J_c\omega_1 = \sum M_C(I^{(e)}) \tag{3}$$

m—The mass of a cudgel
$\omega_1$—The angular velocity of the wrestling cudgel before it hit the ground
$\omega_2$—The angular velocity at the moment of rebound

$J_c$—The rotational inertia of the end rod across the center of mass $C$

As the surface is smooth, the cudgel is only affected by the impact impulse ($I$) in the y direction, $I_x = 0$ ($I_x$ refers as the x direction component impulse), indicated by

$$v'_{Cx} = v_{Cx} \approx \frac{5}{2}\omega_1 l \sin\theta \tag{4}$$

$l$—The length of the tail section

$\theta$—The Angle between the cudgel and the ground at the moment it hits the ground

Selecting the center of mass as the base point, we have:

$$v'_A = v'_C + v'_{AC} \tag{5}$$

$v'_A$—The velocity of endpoint $A$ when it bounces

$v'_C$—The velocity of endpoint $C$ when it bounces

$v'_{AC}$—The difference in velocity between the center of mass $C$ and endpoint $A$

Then projecting along the $Y$-axis, it comes that:

$$v'_{Ay} = v'_{Cy} + \frac{1}{2}\omega_2 l \cos\theta \tag{6}$$

$v'_{Ay}, v'_{Cy}$—The vertical component of the velocity of endpoint $A$ and centroid $C$ at the moment of rebound (i.e., Y direction)

By using the recovery factor as agreed aforementioned,

$$e = \frac{v'_{Ay}}{v_{Ay}} = \frac{v'_{Ay}}{v_2 \cos\theta} = 1 \tag{7}$$

$e$—Recovery Factor

$v_A$—The velocity of endpoint $A$ immediately before the cudgel hits the ground

$v_{Ay}$—The vertical component of the velocity of endpoint $A$ immediately before the club hits the ground

The velocity obtained by an equation of the form:

$$v'_{Ay} = v_2 \cos\theta \tag{8}$$

Substituting (8) into (6) to obtain:

$$v_2 \cos\theta = v'_{Cy} + \frac{1}{2}\omega_2 l \cos\theta \tag{9}$$

By (2) and (3),

$$mv'_{Cy} + m\frac{5}{2}\omega_1 l cos\theta = I \tag{10}$$

$$\frac{1}{12}ml^2(\omega_2 + \omega_1) = I\frac{l}{2}\cos\theta \tag{11}$$

$I$—The impulse of the cudgel hitting the ground

Therefore,

$$v'_{Cy} = \frac{(\omega_2 + \omega_1)l}{6\cos\theta} - \frac{5}{2}\omega_1 l \cos\theta \tag{12}$$

Substituting (12) into (6),

$$\omega_2 = \left\{ \frac{4}{1 - 3(\cos\theta)^2} - 5 \right\}\omega_1 \mu \tag{13}$$

$\mu$—Angular velocity correction factor.

From (13), it is demonstrated that $\omega_2$ are dependent to $\omega_1$ and $\theta$ only. $\omega_2$ is the rebound angular velocity of the bar after the collision, thus it determines the rebound range of the bar. During a slam, if the third section of the cudgel bounce angle is over 90 degrees, the sportsman would be hit. Therefore, the rebound angle should not exceed 90 degrees.

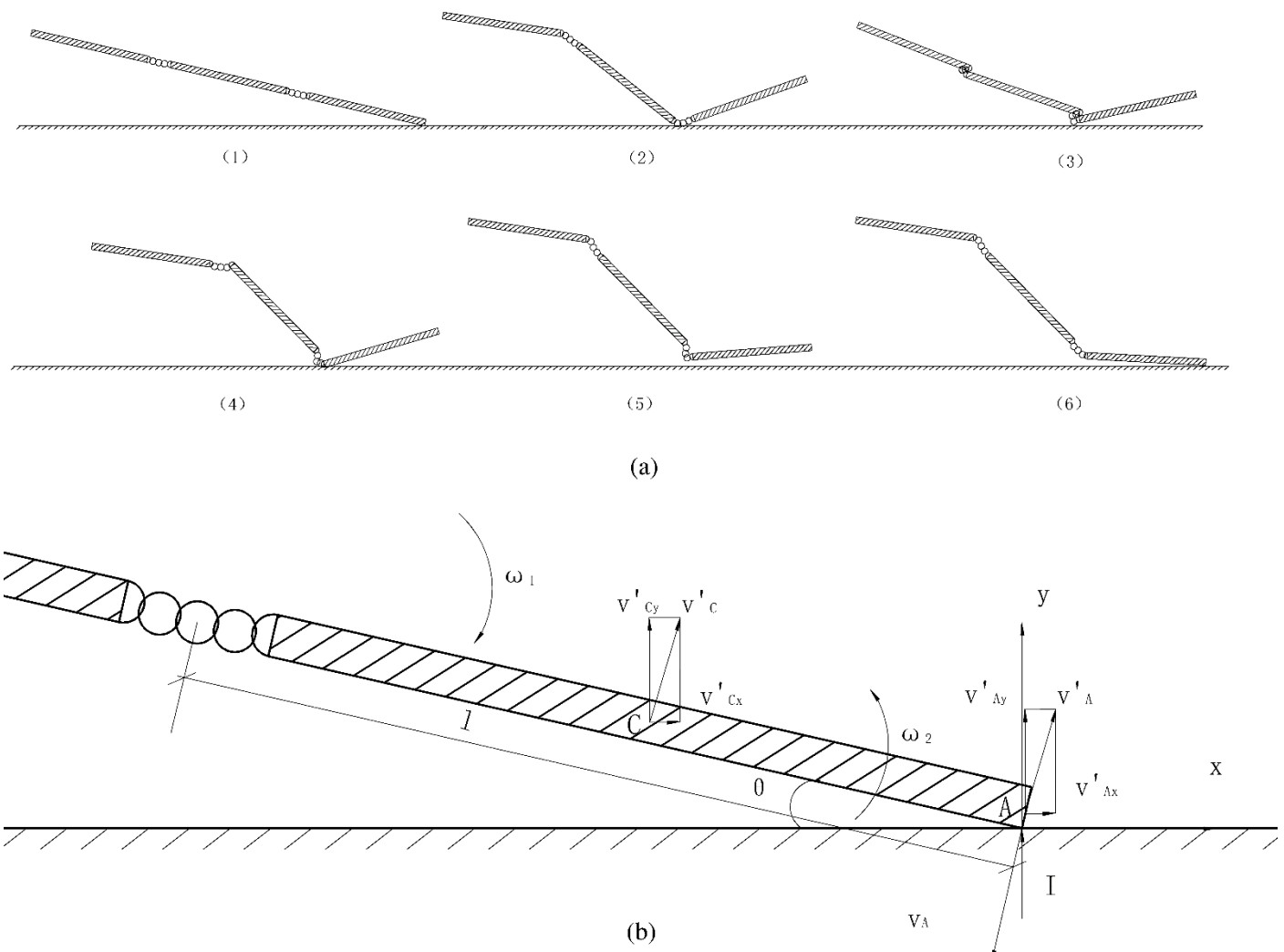

**Figure 3.** Three-section cudgel wrestling process (**a**) diagram and force analysis diagram (**b**).

The critical energy conservation equation for the end of the cudgel when its bounce angle is exactly 90 degrees, it follows [12,13]:

$$\frac{1}{2}m\omega_2^2\left(\frac{l}{2}\right)^2 - 0 = \frac{1}{2}mgl - 0 \tag{14}$$

At the moment,

$$\omega_2 = 2\sqrt{\frac{g}{l}} \tag{15}$$

$g$—free fall acceleration

Therefore, the angular velocity of rebound after a collision should not exceed $2\sqrt{\frac{g}{l}}$, and then we obtained:

$$\left\{\frac{4}{1-3(\cos\theta)^2} - 5\right\}\omega_1\mu \leq 2\sqrt{\frac{g}{l}} \tag{16}$$

According to the results, the arc of the rebound is related to the angle $\theta$ between the cudgel and the ground before the tail section collision and the angular velocity $\omega_1$. $\omega_1$ was mainly related to the speed and strength of wrestling. While wrestling, the arm which holding the cudgel exerting an angular acceleration, and this acceleration would determine the angular velocity $\omega$ at the time of the collision. According (13), it is clear that $\omega_2$ is proportional to $\omega_1$ with a constant ratio, and more bounces back is possibly happening. Actually, the angle $\theta$ between the cudgel and the ground before the collision relates to the height h of the hand when the cudgel wrestled, that is h = I sin $\theta$ (Figure 4). From this point, the higher the height of the hand and the greater the included angle $\theta$, the smaller the $\omega_2$ can be obtained from (13), in turn the greater the arc of the cudgel's rebound. However, the height of the hand should not be as high as possible. If the hand is too high, the cudgel would be restricted by the hand, resulting in the inability to strike with strength. Equation (16) indicated that as long as $\omega_2$ less than $2\sqrt{\frac{g}{l}}$, the bounce of the cudgel is still acceptable. At the same time, $\theta$ also affects the bounce of the cudgel as a variable and the range of $\theta$ is between 0-90 degrees. By (16), under the condition of invariable in $\omega_1$, increasing $\theta$ makes $\omega_2$ decrease and the rebound effect will decrease.

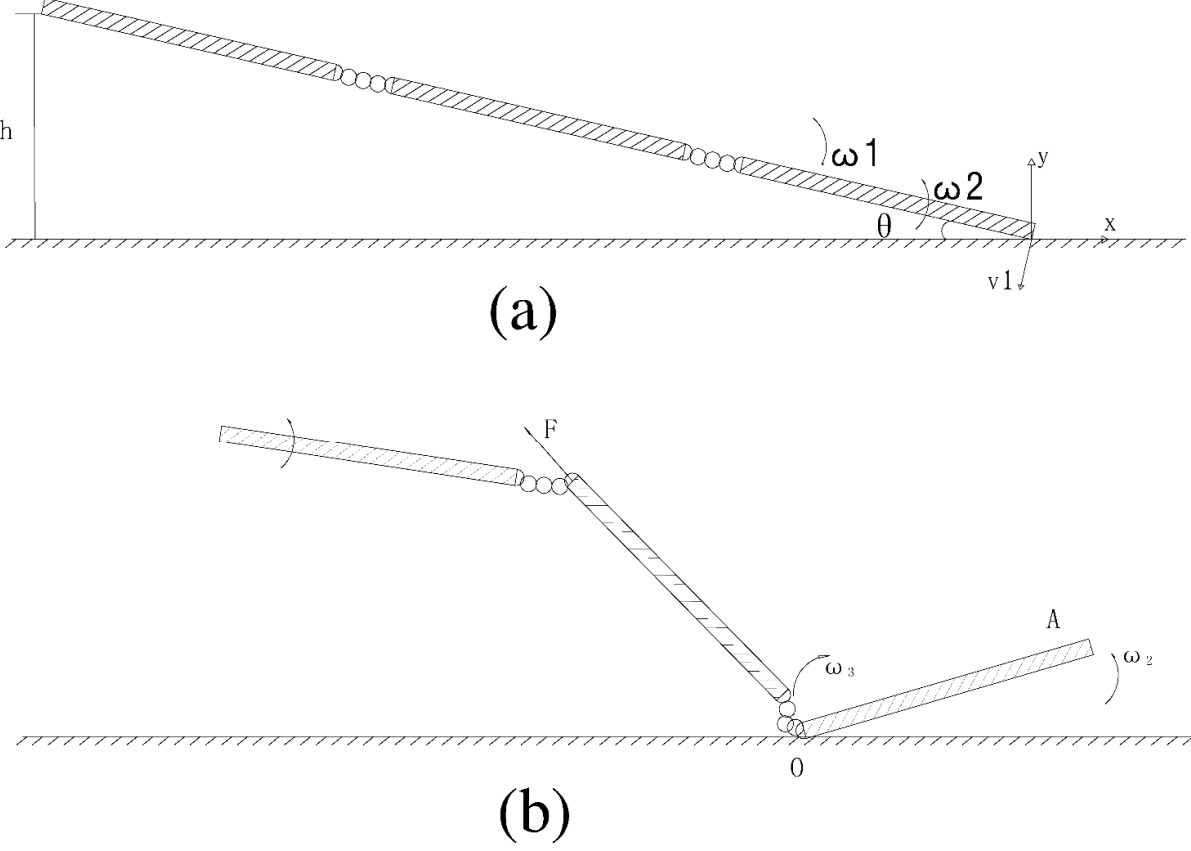

**Figure 4.** Diagram of the moment of three-section cudgel wrestling (**a**) and retrieving (**b**).

On the rebound, the practitioner could change the current state by hand control, giving it an upward force or reverse rotation. It created a force at the center of rotation O in an obliquely upward direction when the force is transmitted to the end, generating a moment in the opposite direction and to be a reverse angular acceleration. It would reduce $\omega_2$ immediately to restraint bounce back of the tail part.

Obviously, the effect of resistance was ignored in the mechanical analysis, as well as the mechanical resistance between the hinges and the cudgel. In reality, resistance played a role, and its rebound was somewhat smaller than the theoretical calculation [14].

### 2.2.2. Theoretical Analysis of Impact Impulse of Three-Section Cudgel Wrestling

When striking on the ground, the impact point usually occurs at the end of the third section, while the impact point is variable when striking an object or a person. Only when a practitioner mastered the hitting technique, he could fully exert the power of the three-section cudgel. Because of the force difference at different impact points, the vibration of the cudgel is completely different.

According to the impact impulse theory in theoretical mechanics [15], when the external impact impulse acts on the impact center in the symmetric plane of the object mass and is perpendicular to the line between the bearing center and the center of mass, it will not cause impact impulse at the bearing [16]. In other words, when striking something with a cudgel and the point struck is exactly the center of percussion, there will be no impact on the hand. If the impact is not in the center of the impact, the hand will feel an impact. For a rigid body rotating on an endpoint, assuming that the external impact impulse I acts on the symmetry plane of the object and the direction of action is perpendicular to the line from the center of rotation to the center of mass, the distance L between the impact center and the axis of rotation can be calculated by the following formula:

$$l = \frac{J_z}{ma} \tag{17}$$

$J_z$—The moment of inertia of the rigid body with respect to the axis of rotation
$a$—he distance from the center of mass to the axis of rotation
$l$—The distance between the center of collision and the axis of rotation
$m$—Mass

Assuming that the third section of the three-section cudgel is a homogenous rod with the mass of m, length of $l$. One end of the cudgel is linked to the second section with a buckle, which was shown in Figure 5.

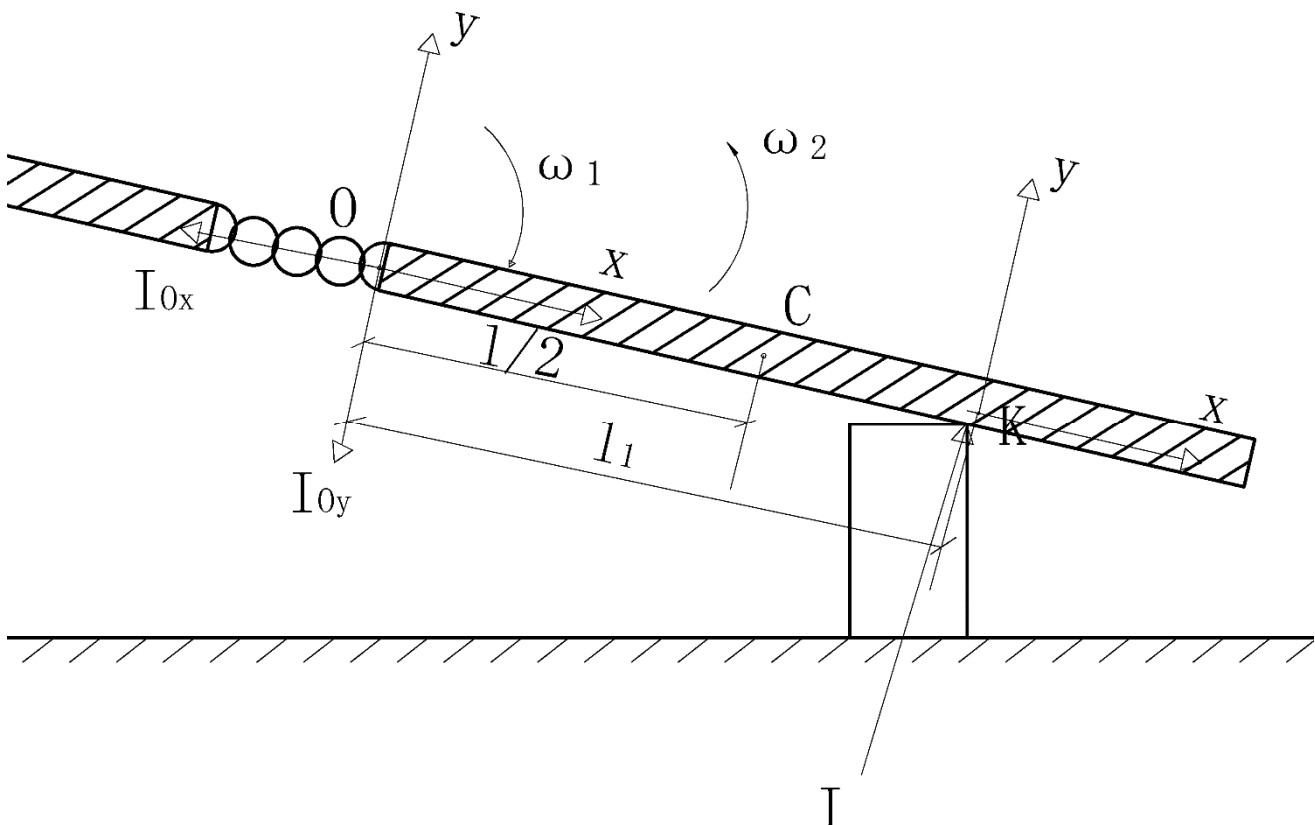

**Figure 5.** Schematic diagram of force analysis of three-section cudgel last section stick striking fixed object.

The tail section of the cudgel impinged on a fixed object at the instantaneous angular velocity of $\omega_2$. Let the recovery factor be e. Supposing that at the beginning and end of the collision, the angular velocity of the cudgel is $\omega_1$ and $\omega_2$, respectively. If the distance between the impact point and the hinge of the buckle is $l_1$, then the recovery factor can be determined.

$$\text{e} = \frac{l_1\omega_2}{l_1\omega_1} = \frac{\omega_2}{\omega_1} \tag{18}$$

$l_1$—The distance from O to the point of impact K

$\omega_1, \omega_2$—The angular velocity of the cudgel at the moment before impact and at the moment of rebound

The impulse moment theorem of hinge point O is:

$$J\omega_2 + J\omega_1 = Il_1 \tag{19}$$

$J$—The rotational inertia of the tail section at O

Thus, the impact impulse can be achieved:

$$I = \frac{J}{l_1}(\omega_2 + \omega_1) = \frac{ml^2}{3l_1}(1+e)\omega_1 \tag{20}$$

The impulse theorem indicates that:

$$\text{m}\left(-\omega_2\frac{l}{2} - \omega_1\frac{l}{2}\right) = I_{oy} - I, I_{ox} = 0 \tag{21}$$

$I_{ox}$—The x direction component impulse of the cudgel at O (parallel to the rod body)

$I_{oy}$—The y direction component impulse of the cudgel at O (parallel to the rod body)

Therefore,

$$I_{oy} = \text{m}\frac{l}{2}(\omega_1 + \omega_2) + I = (1+e)m\left(\frac{l}{3l_1} - \frac{1}{2}\right)l\omega_1 \tag{22}$$

When $I_{oy}$ and hit the center of impact, $\left(\frac{l}{3l_1} - \frac{1}{2}\right) = 0$, that was $l_1 = \frac{2}{3}l$. Therefore, for a section of the cudgel, the best point to strike locates at one-third of the length of the cudgel, where the impact of the opponent of the cudgel is minimal. Therefore, the best hitting position for the third section of a three-section cudgel is one-third of the length of the cudgel, where the impact force of the opponent's cudgel is the least. Thus, the center of impact of the three-section cudgel should be located at a third of the cudgel length from the tip of the cudgel.

### 2.3. Mechanical Analysis of Retrieving Three-Section Cudgel

In the three-section cudgel competition, cudgel withdrawing with a single hand is another difficult movement. This action is a cohesive action after the completion of the three-in-a-row slamming action. The purpose of which is to withdraw the second and third sections of the cudgel around the waist. After completing cudgel slamming, slowly change the stance into the horse stance, and then drag the three-section cudgel from left to right with the right hand (Steps (1) and (2) are shown in Figure 6a below). Then change the horse stance into a bow stance, and the three-section cudgel is wrestled upward by a right hand. The third rod leave the ground (Steps (3) and (4)). The second and third sections will rotate around the two hinges (rings) (Steps (5) and (6)). When the three-section cudgel turns to the position shown in Step (7), quickly grab it with the right hand, and hold it around the waist for a perfect ending. This action seemed simple, but in fact it requires certain skills to catch the returning cudgel, or else the practitioner can get himself hit instead. When wrestling the cudgel, the intensity should be moderate, and all sections of three-sectioned cudgels should be always maintained perpendicular to the ground. When catching the cudgel, the practitioner should be precise to find the right time. Releasing the grasp too early would lead to torsion of the cudgel, thus fail to maintain all three sections on the same

vertical plane. If the cudgel grasping is too late, the third cudgel will hit the second cudgel, causing a violent rebound, and it will not be able to complete the cudgel withdrawing.

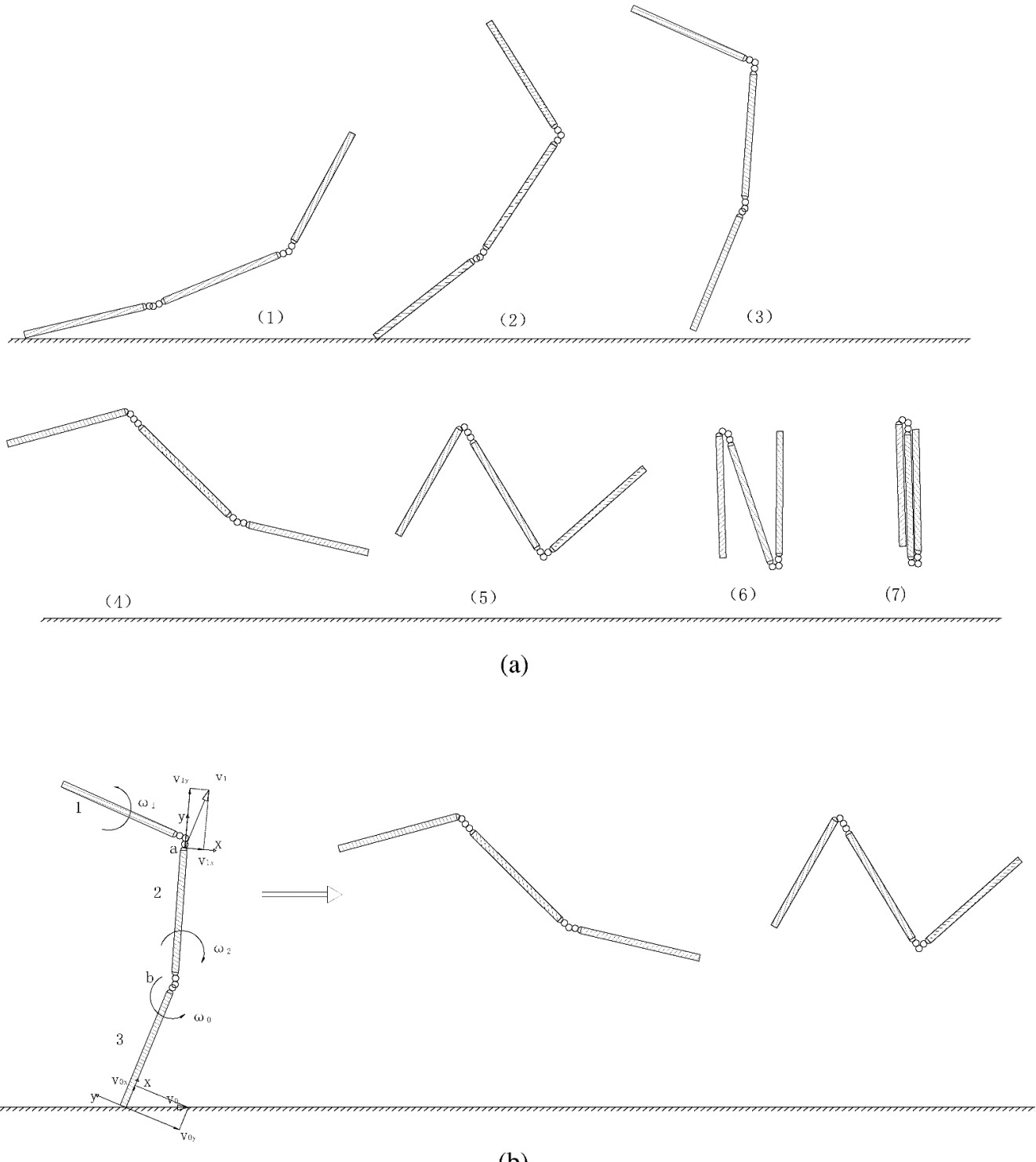

**Figure 6.** Diagram of the process of three-section cudgel retrieving (**a**) and the plane movement process of the second stage of cudgel retrieving (**b**).

The mechanism of the three-section cudgel retrieving could be divided into three stages. In the first stage, a force from left to right was applied to the three sections of the cudgel, so that the three-section cudgel had an initial horizontal velocity $v_0$ to the

right at the tip of the cudgel when the third section of the cudgel was off the ground (as shown in the Figure 6b). In the second stage, the tip of the third section of the cudgel was lifted off the ground by turning the first section counterclockwise. At this point, the initial angular velocity of the first section of the cudgel was $\omega_1$, and the velocity of hinge a between the first and second sections of the cudgel is $v_1$. In the third stage, the first section cudgel rotated vertically and stopped. The second and third sections cudgel rotated horizontally to be withdrawn. Assuming that the three-section cudgel was composed of three homogeneous rods of equal length. Each of which had length *l* with a mass of m. The velocity was decomposed to the axial and radial direction of the corresponding rod, then:

$$v_1 = \omega_1 l \tag{23}$$

$$v_0 = \sqrt{v_{0x}^2 + v_{0y}^2}, v_1 = \sqrt{v_{1x}^2 + v_{1y}^2} \tag{24}$$

$$\omega_0 = \frac{v_{0y}}{l}, \omega_2 = \frac{v_{1x}}{l} \tag{25}$$

$\omega_0$—The angular velocity of the tail section at the b hinge
$\omega_1$—The initial angular velocity of the root section
$\omega_2$—The angular velocity of the middle section at the b hinge
$v_0$—Initial velocity of the root section to the right
$v_1$—The velocity of hinge a between the first and second rods
$v_{0x}, v_{0y}, v_{1x}, v_{1y}$—Components in $v_0$'s and $v_1$'s vertical coordinates (The x direction is along the body of the cudgel, the y direction is perpendicular to the cudgel)

As can be seen in Figure 6b, $\omega_0$ and $\omega_2$ act together on hinge b between the second and third section of the cudgel in opposite directions. As the angular velocity of hinge 2 was slowed, the second section of the cudgel had a small rotation range. Thus, the third section cudgel rotated counterclockwise and upward. Because of its weight, the angular velocity $\omega_0$ was decreasing. Afterwards, the third section of the cudgel quickly rotated in a counterclockwise direction around hinged b. The second section, under its own gravity, decelerated firstly in a counterclockwise direction and then accelerated in a clockwise direction towards the first section of the cudgel until the second and third sections cudgel were folded at the same time to the position of the first section. At this point, the movement was completed.

According to our findings, the key to the cudgel withdrawing action is to control the initial velocity $v_0$ of the tip of the third section and the initial angular velocity $\omega_1$ of the first section in the second stage, so that the second and third sections of the cudgel could be simultaneously rotated to the position horizontally with the first section. In order to reduce the force of the rebound, the second and third sections of the cudgel should be rotated to the overlapping position with the first section at the same time, and the angular velocity of the third section of the rod should be exactly zero in an ideal state. However, the process was actually very fast, thus it was acceptable just to catch the cudgel before it bounces. The higher the initial speed $v_0$ was, the faster the rotation speed of the third section of the cudgel was, and the faster the collecting speed was. When the rotating speed of the third section of the cudgel was relatively high, it would rotate to the vertical direction before the second section. If this happened, a clockwise moment should be applied to the second section through the first section of the cudgel (that is, the first section of the cudgel rotated clockwise at a certain angle) to accelerate the speed of the second section of the cudgel, so that the second and third sections of the cudgel overlapped with the first section at the same time.

### 2.4. ABAQUS Simulation Analysis of Three-Section Cudgels

Cudgel-wrestling is a large displacement rigid body movement in the three-section cudgel routine of wushu [17]. It has fast movement speed, short contact time with the ground and short loading time. Since the load itself is also dynamic, the inertia of the movement structure cannot be ignored in the analysis of the movement of the three-section

cudgel. So, the movement of the three-section cudgel should be analyzed dynamically. In the analysis of rigid body motion, ABAQUS software is usually used to simulate the dynamic changes of objects, because ABAQUS software can more truly reflect the motion of objects, force conditions and some complex contact problems in the process of dynamic rigid body simulation [18]. Therefore, ABAQUS software is used to simulate the movement track of the three-section cudgel in the process of throwing the cudgel, thus further verifying whether it is feasible to study the force situation of the three-section cudgel by using the basic principle of dynamics.

### 2.4.1. Explicit Dynamic Analysis Principle of ABAQUS

The main methods of dynamic analysis in ABAQUS fell into two broad categories: modal superposition procedure [19] and direct-solution dynamic analysis procedure [20]. The direct-solution dynamic analysis procedure is mainly used to solve nonlinear dynamic problems, while the modal superposition is suitable for solving linear dynamic problems. In this case, due to the motion and force of the cudgel during the wrestling movement are complex and the motion is nonlinear. Thence, the optimal method for this case is direct-solution dynamic analysis method. The direct solution method also includes implicit dynamic analysis, subspace-based explicit dynamic analysis, explicit dynamic analysis, direct-solution steady-state dynamic analysis and subspace-based steady-state dynamic analysis. Both explicit dynamic analysis and implicit dynamic analysis can analyze nonlinear dynamic problems.

The explicit dynamic analysis uses ABAQUS/Explicit to solve nonlinear dynamic problems by Explicit direct integration [21], while implicit dynamic analysis uses ABAQUS/ Standard to analyze transient responses of strongly nonlinear problems by implicit direct integration. Both can analyze multiple types of problems. Generally speaking, ABAQUS/Standard is more suitable for smooth nonlinear problems, while ABAQUS/Explicit is more effective in solving complex nonlinear dynamics problems, especially for simulating instantaneous and transient dynamic events, such as explosion and shock problems [22]. For some complex contact problems, ABAQUS/Standard requires many iterations and may be difficult to converge, while ABAQUS/Explicit have shown obvious advantages in saving computing time. In general, ABAQUS/Explicit ability to analyze complex contact problems is superior to ABAQUS/Standard [23]. Therefore, ABAQUS/Explicit is more appropriate in combination with the complicated motion state of the cudgel in the process of wrestling the three-section cudgel.

### 2.4.2. Finite Element Explicit Dynamic Analysis Method of ABAQUS

ABAQUS/Explicit central difference method was used to perform explicit time integral for the equation of motion and the next incremental step was calculated by using the dynamic conditions of one incremental step [24]. At the beginning of the increment step, the program solved the kinematic equilibrium equation, which expressed as mass $M$ of the node times acceleration $\ddot{u}$ equals to the resultant force $(P - I)$ (External force $P$, internal force $I$).

$$M\ddot{u} = P - I \tag{26}$$

$M$—the mass of node
$P$—external force
$I$—internal force
$\ddot{u}$—accelerated velocity
At the beginning of the current increment step (time was $t$), the acceleration was:

$$\ddot{u}\Big|_{(t)} = (M)^{-1} \cdot (P - I)\Big|_{(t)} \tag{27}$$

$t$—time
The central difference method was adopted for the acceleration in time, and the acceleration was assumed to be constant when calculating the change of velocity [25,26]. Apply

this change in velocity pulsed the velocity at the midpoint of the previous incremental step to determine the velocity at the midpoint of the current incremental step:

$$\dot{u}\Big|_{(t+\frac{\Delta t}{2})} = \dot{u}\Big|_{(t-\frac{\Delta t}{2})} + \frac{\Delta t|_{(t+\Delta t)} + \Delta t|_{(t)}}{2}\ddot{u}\Big|_{(t)} \tag{28}$$

$\Delta t$—time increment

The integral of the velocity over time plus the displacement at the beginning of the incremental step to get the displacement at the end of the incremental step.

Thus, the acceleration satisfying the dynamic equilibrium condition was provided at the beginning of the incremental step. With incremental steps, the velocity and displacement were pushed forward "explicitly" in time. It could be seen that the velocity and displacement at the end of the increment step depended only on the acceleration, velocity, and displacement at the beginning. To get accurate results, the time increment should be as small as possible.

The methods of explicit dynamics were summarized as follows:

(1)  Node Calculation

a. Dynamic equilibrium equation

$$u\Big|_{(t+\Delta t)} = u\Big|_{(t)} + \Delta t\Big|_{(t+\Delta t)} \cdot \dot{u}\Big|_{(t+\frac{\Delta t}{2})} \tag{29}$$

$$M\ddot{u} = P - I \tag{30}$$

b. Explicit time integration

$$\ddot{u}\Big|_{(t)} = (M)^{-1} \cdot (P - I)\Big|_{(t)} \tag{31}$$

$$\dot{u}\Big|_{(t+\frac{\Delta t}{2})} = \dot{u}\Big|_{(t-\frac{\Delta t}{2})} + \frac{\Delta t|_{(t+\Delta t)} + \Delta t|_{(t)}}{2}\ddot{u}\Big|_{(t)} \tag{32}$$

$$u\Big|_{(t+\Delta t)} = u\Big|_{(t)} + \Delta t\Big|_{(t+\Delta t)} \cdot \dot{u}\Big|_{(t+\frac{\Delta t}{2})} \tag{33}$$

$\dot{u}$—velocity
$u$—displacement

(2)  Unit computation

a. Calculate the strain increment of the unit $d\varepsilon$ according to strain rate
b. Calculate the stress $\sigma$ according to the constitutive relation

$$\sigma|_{(t+\Delta t)} = f(.\sigma|_{(t)}, d\varepsilon.) \tag{34}$$

c. Integrated node internal forces $I |_{(t+\Delta t)}$
d. Set time t to be $t + \Delta t$ and repeat step $d$.

## 3. Results

*3.1. The Influence of the Height between the Cudgel Holder and the Ground for the Cudgel Wrestling*

The models (cudgel and ground) parameters were shown in Table 1. In consideration of the contact between the three-section cudgel and the ground, the tangential direction was friction contact (the friction coefficient was 0.1) and the normal direction was hard to contact. The height of the hand H and the initial angular velocity $\omega$ of the cudgel were the two main factors to be considered in the wrestling process. The software ABAQUS is used to simulate the three-section cudgel throwing action. By changing the angular velocity of the cudgel and the height of the cudgel holder, the rebound height of the cudgel after the drop is calculated when the angular velocity and the height of the cudgel holder are different.

**Table 1.** Model material parameters.

| Structure | Elasticity Modulus/(Gpa) | Poisson Ratio | $\rho/(\text{g/cm}^3)$ | Friction Coefficient (Tangential Direction) |
|---|---|---|---|---|
| The three-section cudgel (The whole) | 2.1 | 0.3 | 7.85 | 0.1 |
| Ground | 2.1 | 0.3 | 7.85 | 0.1 |

Assuming that when the cudgel was wrestled, at a horizontal position, the overall angular velocity of the staff $\omega$ = 60 rad/s, the height of the hand was at H. By changing H variable, we analyze how the height of the hand affect the rebound of the end of the cudgel after it hit the ground. The height of the hand was set to be H = 500 mm, 600 mm, 700 mm, 800 mm, 900 mm and 1000 mm, respectively. The movements of the staff during a full wrestle were simulated using the software as Figure 7 and the corresponding results were shown in Table 2.

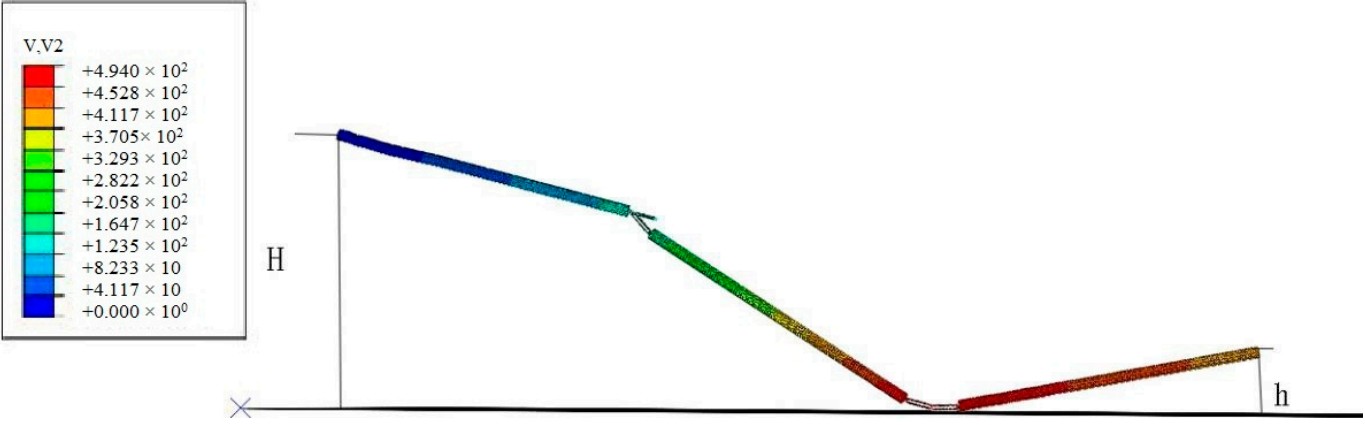

**Figure 7.** The maximum displacement cloud map of the rebound after the cudgel was wrestled simulated by ABAQUS.

**Table 2.** The height of the hand corresponding to the height of the rebound at the node of the rod tip.

| The height of the hand when wrestling a cudgel (H/mm) | 500 | 600 | 700 | 800 | 900 |
|---|---|---|---|---|---|
| Rebound height (h/mm) | 135.324 | 236.867 | 45.104 | 82.451 | 174.404 |

After analyzing the result, the relationship between the height of the hand and the rebound height of the end of the staff was shown in Figure 8. By analyzing the graph, we would notice that, when the value of H was between 70 cm to 80 cm, the maximum height of the rebound was the smallest. According to the practice experience and the analysis, we found that the height of the hand before the collision of the third section was about 1.5 times the length of one cudgel section. When the height between the cudgel holder and the ground is greater than 100 cm or less than 55 cm, the rebound height will also be reduced, but the horizontal displacement generated at this time is very large, which will lead to the poor effect of throwing the cudgel. Therefore, in the process of throwing the cudgel, the best effect is achieved when the height of the cudgel holding hand is between 70 cm and 80 cm from the ground.

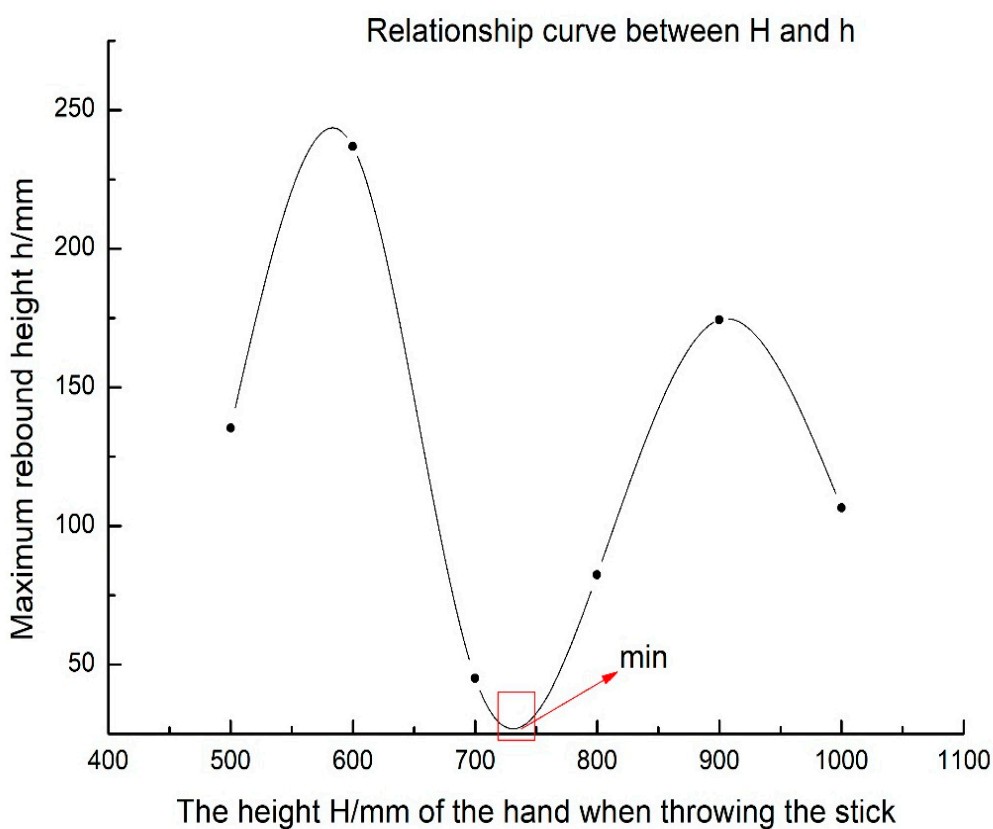

**Figure 8.** The relation graph between the height H of the hand and the height h of the maximum rebound of the cudgel tip.

### 3.2. The Influence of Initial Angular Velocity of the Three-Section Cudgel for the Cudgel Dropping

At the process of a wrestling, we defined H as the height of the hand, by the value of H (H = 800 mm). We observed that how initial angular velocity affect the rebound of the cudgel. The initial angular velocity of the three-section cudgel was set as $\omega$ = 60 rad/s, 70 rad/s, 80 rad/s, 90 rad/s, 100 rad/s, 110 rad/s and 120 rad/s, respectively. The simulation results using ABAQUS software are shown in Figure 9.

By analyzing Figure 10, we noticed that the maximum height of the rebound was very similar with different initial angular velocities, at 4.5–9 cm. Hence, the initial angular velocity of the three-section cudgel might not actually affect the height of the rebound. Moreover, we analyzed horizontal movement of staff at 8 s with different initial angular velocities as shown in Figure 9.

As was shown in Figure 10, the initial angular velocities caused significant impact to the horizontal movement of cudgel at 8 s. By excluding the errors, the horizontal movement of cudgel increased approximately linearly with the increase of the initial angular velocity.

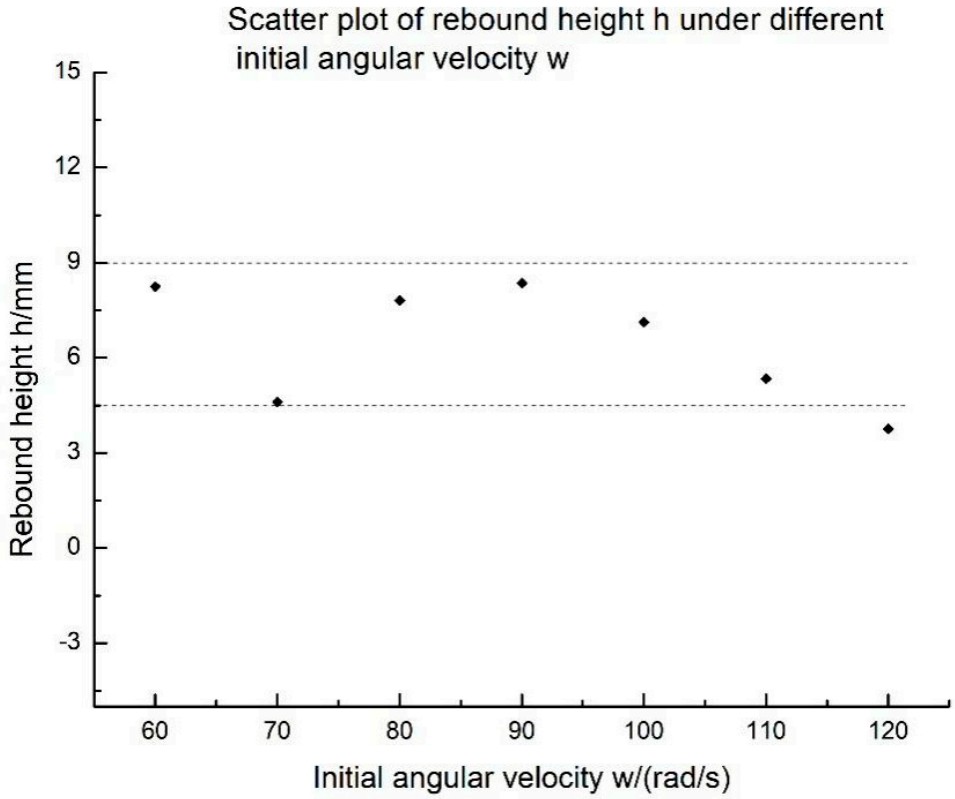

**Figure 9.** Scatter diagram of the rebound height h at different initial angular velocity w of three-section cudgel.

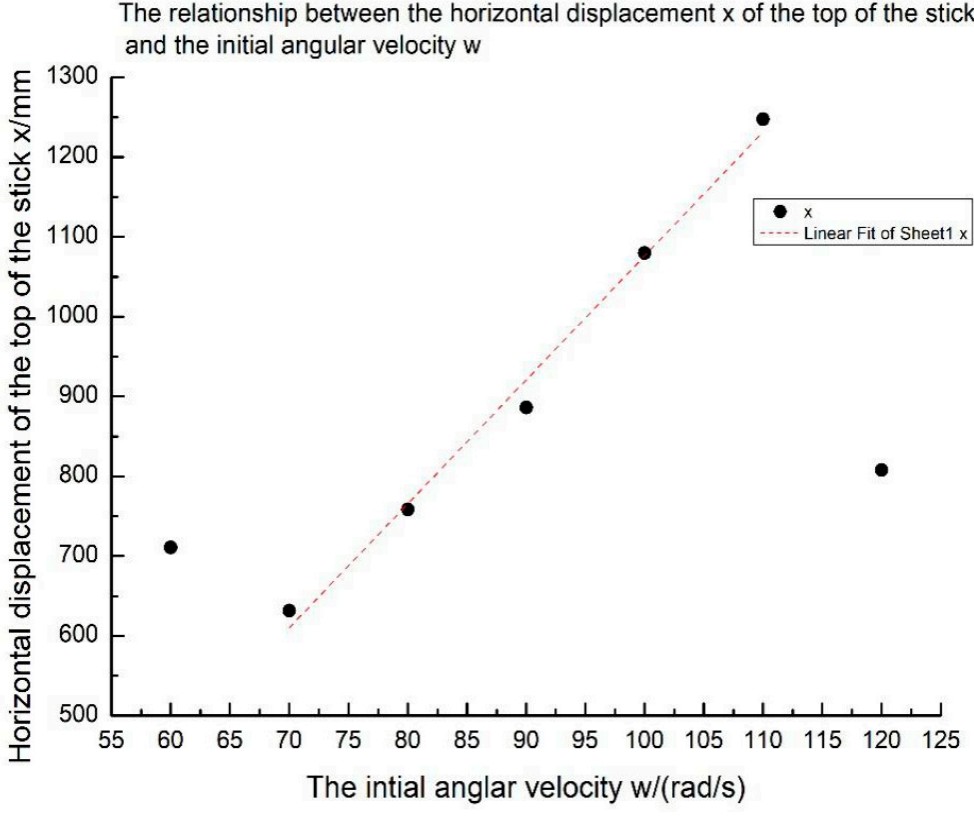

**Figure 10.** Scatter diagram of horizontal displacement at the top of the lower rod with different initial angular velocities.

## 4. Discussion

This paper analyzed the rebound process and the force when the cudgel hits the object from the theoretical mechanics. To simplify the calculation, the influence of air resistance, friction resistance and mechanical action on the three-section cudgel was ignored. When the height between the cudgel holder and the ground was about 1.5 times the length of the middle section, the rebound height of the cudgel was the smallest. When the cudgel strokes an object, the vibration and rebound of the cudgel were minimal and the impact of the opponent was minimal when the point of impact between the cudgel and the object was in the center of impact.

From the perspective of mechanics analysis of the three-section cudgel's movement, the key to the movement of the three-section cudgel retrieving was to control the initial velocity $v_0$ of the third cudgel's cudgel tip and the initial angular velocity $\omega_1$ of the first cudgel in the second stage and turn the second and third bar to overlap the first bar. The greater the initial speed $v_0$, the faster the third cudgel rotated and retrieved.

The software ABAQUS was used to simulate the action of wrestling cudgel in the three-section cudgel routine of wushu. Firstly, software simulation analysis and calculation of the three-section cudgel displacement cloud images when the right hand held the three-section cudgel at different heights of the first bar, it was found that the height of the right hand and the initial angular velocity when the three-section cudgel turning to the horizontal level had a great influence on the effect of wrestling the cudgel. Among them, the height of the hand had a great influence on the rebound height of the wrestling cudgel. For the size of the three-section cudgel in the study, the best height of the hand was 70–80 cm. At this time, the rebound height of the cudgel was the lowest and the action link after the wrestling cudgel could be better controlled. Secondly, the initial angular velocity had no obvious effect on the bounce height of the three-section cudgel, but it did have some effect on the horizontal displacement of the cudgel. With the increase of the angular velocity, the horizontal displacement of the cudgel also increases. Compared with the effect of rebound, the effect of horizontal displacement could be ignored in the process of controlling the falling cudgel, and the horizontal displacement was easy to be controlled. So, the main factor that affected cudgel wrestling was the height of the hand. In addition, in the process of software simulation, it was easy to see that when the three-section cudgel fell to the ground, not always the tip of the cudgel touched the ground first, sometimes the whole tail section fell to the ground at the same time, sometimes the bottom of the tail section and the joint of the hinge touched the ground first, and the rebound was the least when the whole tail section fell to the ground at the same time.

The height of the first section of the three-section cudgel held by the right hand simulated by ABAQUS software was similar to the height of 1.5 times the length of a section of the cudgel in the theoretical mechanics analysis [27]. Therefore, the results of this study had credibility and operability.

However, there were several limitations to this study. This study is based on the assumption of an ideal rigid body, which is a solid of finite size that can ignore deformation. Inside the rigid body, the distance between the points does not change, whether or not there is an external force. In practice, however, the ideal rigid body does not exist [28,29]. In addition, when analyzing the collision process, the impact between three cudgels and the ground and objects will actually produce a little plastic deformation which is actually not accord with perfectly elastic collision situation. Therefore, there is still room for improvement the accuracy especially in the analysis of the interaction between the three short cudgels of the three-section cudgel and the influence of the external environment on the three-section cudgel [30]. The completion of other movements in the three-section cudgel method also needs to be further studied in the future work.

## 5. Conclusions

In summary, based on the results of mechanical theory analysis, we put forward the following three training key points of three-section cudgel movement. Firstly, the

three-section cudgel should be kept in the same vertical plane perpendicular to the ground in the whole process of retrieving the cudgel. Secondly, when the speed is fast, a clockwise moment should be applied to the second bar through the first bar. Finally, cudgel retrieving in due course, be rapid and accurate. For beginners, it is recommended to take the cudgel with both hands first.

**Author Contributions:** Software, X.H.; validation, Y.J., M.X. and J.Y.; formal analysis, X.H. and Y.J.; resources, M.X.; writing—original draft preparation, M.X., Y.J. and X.H.; writing—review and editing, M.X., Y.J. and Y.G.; visualization, Y.J. and X.H.; project administration, M.X.; supervision, Y.G. All authors have read and agreed to the published version of the manuscript.

**Funding:** This research received no external funding.

**Conflicts of Interest:** The authors declare no conflict of interest.

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
