# Peer review of "Dynamic Analysis of the Complex Motion of Three-Section Cudgel in Wushu Sports"

_applsci, doi:10.3390/app112110407_

Round 1

Reviewer 1 Report

The study addresses the complex mechanical relationships between a three sectional staff (cudgel) and its impact with the ground. The research analyses both the performance of the striking movement and the safety of its rebound and retrieval. It is well written with only minor improvements in literacy and sentence structure required. The authors do well to explain  some of the concepts in practical terms. Given the novel nature of this sport, the level of interest to readers of the Journal of Applied Sciences is unknown. This was evidenced by the difficulty in finding certain aspects of the skill in general internet searches. However, the paper does provide adequate references that relate to the movement pattern.

Whilst the paper includes substantial mechanical detail in the form of theoretical analysis, it would be beneficial to include support for the calculations and formulae via high speed motion analysis. Given the complexity of a three segmented impact, video supporting each phase for the various parameters would add confidence to the results presented. This would lengthen the paper substantially and may require the study to be simplified or submitted as two separate papers.

Other than the above, I only have a couple of specific comments to improve the paper. Some of the formulae are introduced prior to defining certain variables. For example, c and v on page 3. An example of where variables are defined well is on page 6 for equation 17. Having the definitions prior to their use helps the reader understand the relationships presented in each equation. Secondly, some of the text used within the diagrams needs to be made clearer (e.g. Figures 3 to 6).

Author Response

We deeply appreciate the helpful suggestions from the editors and reviewers. We believe our manuscript benefited significantly from this revision. Based on the instructions given, we prepared the point-by-point response and have made relevant revisions in the manuscript with tracked changes. The main revisions in the article and their corresponding responses are shown in the attachment.

Reviewer 2 Report

The research is appropriately planned. I rate this article as positive with minor corrections.  

The abstract must be written in a clearer and defined way (results).

In the theoretical part, I find the characteristics of the device very deficient. Chapters methods and results are written very well.

In conclusion, we would expect a more in detail explained study limitations.

Author Response

We deeply appreciate the helpful suggestions from the editors and reviewers. We believe our manuscript benefited significantly from this revision. Based on the instructions given, we prepared the point-by-point response and have made relevant revisions in the manuscript with tracked changes. The main revisions in the article and their corresponding responses are shown in the attachement.
